# The Role of Exosomes in Breast Cancer Diagnosis

**DOI:** 10.3390/biomedicines9030312

**Published:** 2021-03-18

**Authors:** Claudia Piombino, Ilenia Mastrolia, Claudia Omarini, Olivia Candini, Massimo Dominici, Federico Piacentini, Angela Toss

**Affiliations:** 1Department of Oncology and Hematology, Azienda Ospedaliero Universitaria di Modena, 41124 Modena, Italy; claudia.piombino@outlook.com (C.P.); claudia.omarini@gmail.com (C.O.); massimo.dominici@unimore.it (M.D.); federico.piacentini@unimore.it (F.P.); 2Laboratory of Cellular Therapy, Department of Medical and Surgical Sciences, University of Modena and Reggio Emilia, 41124 Modena, Italy; 3Rigenerand srl, Medolla, 41036 Modena, Italy; olivia.candini@rigenerand.it; 4Division of Oncology, Department of Medical and Surgical Sciences, University of Modena and Reggio Emilia, 41124 Modena, Italy; 5Division of Oncology, Department of Surgery, Medicine, Dentistry and Morphological Sciences, University of Modena and Reggio Emilia, 41124 Modena, Italy

**Keywords:** exosome, liquid biopsy, breast cancer, diagnosis, BRCA, precision oncology, miRNA, vesicles

## Abstract

The importance of molecular re-characterization of metastatic disease with the purpose of monitoring tumor evolution has been acknowledged in numerous clinical guidelines for the management of advanced malignancies. In this context, an attractive alternative to overcome the limitations of repeated tissue sampling is represented by the analysis of peripheral blood samples as a ‘liquid biopsy’. In recent years, liquid biopsies have been studied for the early diagnosis of cancer, the monitoring of tumor burden, tumor heterogeneity and the emergence of molecular resistance, along with the detection of minimal residual disease. Interestingly, liquid biopsy consents the analysis of circulating tumor cells, circulating tumor DNA and extracellular vesicles (EVs). In particular, EVs play a crucial role in cell communication, carrying transmembrane and nonmembrane proteins, as well as metabolites, lipids and nucleic acids. Of all EVs, exosomes mirror the biological fingerprints of the parental cells from which they originate, and therefore, are considered one of the most promising predictors of early cancer diagnosis and treatment response. The present review discusses current knowledge on the possible applications of exosomes in breast cancer (BC) diagnosis, with a focus on patients at higher risk.

## 1. Liquid Biopsy and Extracellular Vesicles

Cancer is a dynamic and heterogeneous entity following the principles of clonal evolution. Different areas of the same primary tumor show different genomic profiles, while metastases acquire new molecular aberrations compared to primary tumors. Therapy-related biomarkers may change throughout cancer progression ‘in time and space’. As a result, the measurement of the biomarker of interest at multiple time points and different sites of the tumor may provide crucial information for patient management. On these grounds, precision oncology has highlighted the need of providing the most appropriate and effective treatment to each cancer patient, assuming that inter- and intra-tumor genetic heterogeneity could explain sensitivity or resistance to anticancer agents [1]. The primary goal of precision oncology is, therefore, to discover molecular biomarkers predicting prognosis and response to specific therapies, helping to anticipate the emergence of unexplained drug resistance [2]. Nevertheless, obtaining serial samples of tumor tissue is impractical and complicated by spatial heterogeneity and sampling bias. Indeed, more comprehensive and accessible tumor genome information is needed to provide an accurate account of the whole tumor than that obtained through single biopsy. Interestingly, an attractive alternative to overcome the limitation of repeated tissue sampling is provided by the analysis of peripheral blood samples as ‘liquid biopsy’.

Liquid biopsy is being developed as a promising new technique in the field of precision oncology. It is a minimally invasive prognostic and diagnostic tool that could overcome the limits of surgical biopsy [3]. Blood draws can easily be performed serially. Thus, blood is an ideal compartment for the detection of prognostic and predictive biomarkers. Moreover, liquid biopsy has several potential clinical applications. These include early tumor diagnosis [4,5], the monitoring of tumor burden [6,7,8,9], tumor heterogeneity and the emergence of molecular resistance [10], and the detection of minimal residual disease [6]. In particular, liquid biopsy mainly targets materials pulling away from tumor edges and swept away by the bloodstream, including circulating tumor cells, circulating tumor DNA and extracellular vesicles (EVs) [11]. It is well-known that nucleic acids are present in biological fluids in healthy subjects in stable low concentrations and are immunologically inactive; however, they change dramatically in cancer and autoimmune disorders [12]. The circulating DNA is also internalized in EVs, which protect it from nuclease degradation or recognition as dangerous by immune cells and provide their effective clearance. The features of circulating DNA and its packaging in vesicles reflect the state of cell of origin, such as apoptosis, necrosis, phagocytosis or active secretion [13].

EVs are small lipid bilayer-enclosed vesicles, actively released by all viable cells that play a vital role in cell communication [14]. They carry transmembrane and nonmembrane proteins as well as metabolites, lipids, messenger RNAs, microRNAs, long-noncoding RNA, and DNA [15,16]. In recent years, the interest in EVs has rapidly increased [17] and several studies have demonstrated their potential use as diagnostic, prognostic and therapeutic agents in clinical settings [18]. In 2014, the International Society for Extracellular Vesicles (ISEV) board members provided a list of minimal information regarding EVs, updated in 2018. According to ISEV guidelines, the term EVs includes three types of vesicles, namely exosomes, microvesicles, and apoptotic bodies, based on origin and size of diameter [19]. In detail, exosomes are defined as intra-luminal vesicles with a diameter ranging from 30 to 150 nm derived from the multi-vesicular (MV) bodies, formed by budding of the endosomal membranes and secreted in the extracellular space upon fusion of late endocytic compartments with the plasma membrane. Microvesicles include different populations of vesicles, which are in the nano-range of 50–200 nm, and larger vesicles up to 1 µm, which include the pre-apoptotic vesicles. They are generated by plasma membrane budding and are shed in the extracellular space. Apoptotic bodies, with a diameter ranging from 1 to 5 µm, are a class of vesicles released by cells exclusively during apoptotic cell death and their cargo is mainly enriched with nuclear fragments [20] (Figure 1A). In particular, exosomes are extremely abundant in all biological fluids, including serum, cerebrospinal fluid [21] plasma, saliva, breast milk [22] and urine [23]. When exosomes were discovered in 1983 [24], they were first believed to operate as cellular garbage disposal [25]. Since then, several researchers have investigated their biological roles. These include, but are not limited to, antigen presentation, immune regulation, apoptosis evasion, drug resistance and immune surveillance escape [26,27]. Moreover, exosomes derived from cancer cells have been demonstrated to play a key role in facilitating tumorigenesis by regulating angiogenesis, immunity, and metastasis [28,29] (Figure 1B). By way of example, Peinado et al. [28] observed how melanoma-derived exosomes increase the metastatic behavior of primary tumors by permanently “educating” bone marrow progenitors via the MET receptor. Besides, melanoma-derived exosomes induce vascularization at pre-metastatic sites and reprogram bone marrow progenitors towards a pro-vasculogenic phenotype. Al-Nedawi et al. demonstrated that the transmission of the constitutively active EGFRvIII via EVs not only transfer oncogenic activity among cancer cells but also activates autocrine VEGF signaling in endothelial cells stimulating tumor angiogenesis [30,31]. Finally, another key example of the role of exosomes in metastatization has been shown in pancreatic cancer, where EVs promote pre-metastatic niche formation in the liver through macrophage inhibitory factor signaling and consequent fibrotic liver environment [32].

## 2. Focusing on Exosome Isolation and Characterization

Despite growing interest in this field, our understanding of the biogenesis, release, uptake and function of EVs remains limited. A key limitation to the specific characterization of EV subpopulations has been the technical difficulty in isolating and characterizing pure populations of specific subtypes. This appears to be the case because the methods currently available lead to the systematic co-isolation of EVs of distinct subcellular origins [33].

Within the EV population, exosomes have sparked great scientific interest in recent years. This is due to biological fingerprints practically mirroring those of the parental cells from which they originate [34]. Exosomes are identified primarily by their size, which, however, has not been universally defined. Typically, exosomes are considered as vesicles of 30–150 nm in diameter up to 200 nm. However, based on their biogenesis, recent studies have identified three transmembrane proteins belonging to the tetraspanin family (CD9, CD81 and CD63). These are commonly found in exosomes and are often enriched in the vesicles compared to cell lysate [35] (Figure 1C).

The key role of exosomes in cell-to-cell communication and tumorigenesis has led to their isolation and quantification as major challenges in both basic research and clinical applications [36]. The need for standard exosome characterization methodologies that are reproducible became imperative to allow for the use of exosomes as potential biotools in diagnosis and treatment of various diseases. As the first step toward improving knowledge in this field and developing exosome-based assays, exosomes have to be reliably and efficiently isolated from several body fluids. To date, five groups of exosome isolation techniques have been developed. These are differential ultracentrifugation-based techniques, size-based techniques, immunoaffinity capture-based techniques, exosome precipitation and microfluidics-based techniques [37]. Differential ultracentrifugation or commercial kits allowing for the precipitation of smaller EVs are among the most common methods for isolating exosomes from serum or plasma. Exosome precipitation is easy to use and does not require any specialized equipment, allowing for easy integration into clinical usage [38].

Alongside isolation techniques, the characterization of the exosomes is also critical to developing exosome-based assays. Characterization methods of exosomes are categorized into biophysical and molecular methods. Biophysical methods, of which the most common are nanoparticle tracking analysis (NTA) and transmission electron microscopy (TEM), are normally used to determine the size distribution of exosomes in samples. However, molecular information is not achieved. Molecular methods such as flow cytometry allow for the identification of molecular markers, e.g., surface receptors, membrane proteins present on exosomes. Moreover, since exosomes are of intracellular origin, they are packaged with a bio-macromolecular cargo of DNA, RNA and proteins (Figure 1C). The exosome cargo therefore represents molecular bioprint of the cell-of-origin and is important in initiating or suppressing various signaling pathways in recipient cells, also responsible for metastasis and drug-resistance [36].

## 3. Clinical Applications of Exosome Research

Cancer cells may secrete a larger number of exosomes related to normal cells [39], with a number of cancer-specific biomarkers. As they are easily accessible and stable in vitro, exosomes have been considered to be one of the most promising predictors of early cancer diagnosis and treatment response [40].

Based on increasing awareness of the importance of exosomal content, two test kits based on liquid biopsy approach have been made commercially available since 2016 to detect prostate and lung cancer markers. These are ExoDx^®^ Prostate (IntelliScore) and ExoDx^®^Lung(ALK) [41]. ExoDx^®^ Prostate (IntelliScore) is a urine exosome gene expression assay suitable for men after 50 years of age with a prostate-specific antigen (PSA) of 2–10 ng/mL or PSA in a “gray zone”, considering initial biopsy. ExoDx^®^ Prostate (IntelliScore) returns a risk score that determines patients’ risk of clinically significant prostate cancer (Gleason Score ≥ 7) on prostate biopsy. A score above the validated cut-point of 15.6 is associated with increased likelihood of Gleason Score ≥ 7 prostate cancer on biopsy [42]. Similarly, ExoDx^®^Lung (ALK)—validated in the Exosome Diagnostics CLIA laboratory—isolates and analyzes exosomal RNA contained in blood specimens for the purpose of detecting *EML4-ALK* fusion transcripts in the plasma of lung cancer patients whose primary tumors carry this type of mutation. The ExoDx^®^Lung (ALK) test can be used both at baseline to help guide treatment choice, and longitudinally to display patient progress during therapy [43].

The search for circulating tumor materials is emerging as a novel method for breast cancer (BC) diagnosis as well. Since stage at diagnosis remains the main prognostic factor [44], accurate blood tests matching the sensitivity and specificity of mammographic screening would be helpful in early detection [38]. Against this backdrop, a large number of researchers are studying exosomes due to their potential as highly accessible source of detailed information on tumor biological features (proteins and nucleic acids) obtained through liquid biopsy [45]. The present review discusses current knowledge on the possible applications of exosomes in early BC diagnosis, with a focus on patients at higher lifetime BC risk.

## 4. Exosomal Proteins in Breast Cancer Diagnosis

Proteins located on the surface of, as well as within exosomes, may also be used as cancer biomarkers. As shown by proteomic results available in the ExoCarta and EVPedia databases [46], exosomes exhibit specific protein profiles according to cellular origin. As previously mentioned, tetraspanins are abundantly expressed in exosomes [47]. These are a protein superfamily that interacts with a large variety of transmembrane and cytosolic signaling proteins [48,49]. In particular, tetraspanin CD9, along with metalloprotease ADAM10, heat-shock protein HSP70 and Annexin-1, are general marker proteins detected in serum and pleural effusion-derived exosomes from patients with BC or BC cell lines [50]. Interestingly, Wang and colleagues [51] recently showed that the level of exosomal tetraspanin CD82 was significantly higher in the serum of BC patients compared to healthy controls, while the expression of CD82 significantly increased with malignant breast cancer progression. Furthermore, the combined expression of urinary exosomal tetraspanin CD63 and miR-21 had a 95% sensitivity to early BC detection, although both markers are not specific to BC [52].

Rupp et al. [53] reported that the epithelial cell adhesion molecules EpCAM and CD24 could be used as markers to specifically identify cancer-derived exosomes in ascites and pleural effusions from BC and ovarian cancer. In the same period, Moon and colleagues [54,55] found that both plasma levels of developmental endothelial locus-1 protein (Del-1) and fibronectin expressed by circulating exosomes were significantly higher in patients with BC than in controls. Moreover, they almost returned to normal after tumor removal, proving to be closely related to tumor presence. Additionally, Khan et al. [56] demonstrated that exosomal-Survivin, particularly Survivin-2B, may be employed as a diagnostic and/or prognostic marker in early BC patients.

Interestingly, exosomes from gastric, breast and pancreatic cancer carry members from the human epidermal growth factor receptor (HER) family [57,58,59]. In HER2-overexpressing BC cell lines, HER2-positive exosomes modulate sensitivity to Trastuzumab and, consequently, HER2-driven tumor aggressiveness [59]. Although not specific to early BC diagnosis, HER2 could be a useful biomarker for anticipating drug-resistance during treatment, which represents the principal limiting factor to the development of cures in cancer patients.

Additionally, Melo and colleagues [60] identified a cell surface proteoglycan, glypican-1 (GPC1), specifically enriched on cancer cell-derived exosomes. They observed that GPC1-positive circulating exosomes were specifically and sensitively detectable in the serum of patients with pancreatic cancer. Elevated GPC1 levels have also been observed on exosomes from BC cells, suggesting a possible use of this exosomal biomarker to identify BC early [61].

More recently, Kibria et al. [62] used an automated micro flow cytometer to profile protein expression of exosomes isolated from cell lines and blood of BC patients and healthy controls. They observed a significant reduction in CD47 expression in circulating exosomes from BC patients, compared to controls. Notably, CD47 is a cancer-related surface protein whose expression prevents recognition of cancer cells by the innate immune system, thus facilitating tumor progression [63].

Finally, other studies demonstrated the higher expression of serum exosomal-annexin A2 (exo-AnxA2) in BC patients compared to non-cancer females, especially for triple-negative BC (TNBC) rather than luminal and HER2-positive BC. Besides, high expression of exo-AnxA2 levels in BC was significantly associated with tumor grade, poor overall survival and poor disease-free survival. This study also showed that exo-AnxA2 promotes angiogenesis. Therefore, exo-AnxA2 represents a potential prognostic biomarker and therapeutic target of TNBC [64].

## 5. Exosomal MicroRNAs in Breast Cancer Diagnosis

MicroRNAs (miRNAs) are short, noncoding single-stranded RNAs that regulate gene expression at a post transcriptional level by binding to the 3′ untranslated region of its target mRNA, leading to translational inhibition or mRNA degradation [65]. Exosomes contain plenty of miRNAs [66], and several studies investigated the role of exosomal miRNA expression in mediating biological effects in receiving cells [67,68,69,70,71,72]. In particular, miRNAs stably exist in body fluids by virtue of their packaging in exosomes, which protects them from degradation [73]. Interestingly, exosomal miRNAs can act as novel ideal biomarkers in BC, because their expression profile correlates with tumorigenesis and tumor progression [74,75,76,77].

In 2016, Hannafon et al. [78] showed that the levels of exosomal miR-21 and miR-1246 in plasma were markedly higher in BC patients than in healthy subjects. This suggests their potential use as biomarkers in BC, although miR-21 and miR-1246 are ubiquitous in human exosomes. These data are in keeping with other studies that described high levels of these miRNAs in serum or plasma from BC patients. In detail, Shimomura and colleagues [79] evaluated serum miRNA expression profiles using highly sensitive microarray analysis, discovering a combination of five miRNA (miR-1246, miR-1307-3p, miR-4634, miR-6861-5p and miR-6875-5p) able to detect BC with high sensitivity, specificity and accuracy, even in the case of ductal carcinoma in situ (DCIS). Additionally, Fu et al. [80] found that miR-382-3p and miR-1246 were significantly upregulated in the serum of BC patients, while miR-598-3p and miR-184 were significantly downregulated. Finally, a meta-analysis of Li and colleagues [81] suggested that miR-21 is a potential biomarker for early diagnosis, with high sensitivity and specificity being significantly up-regulated in BC.

Although miR-145, miR-155, and miR-382 have been proposed as non-invasive biomarkers to distinguish BC patients from healthy individuals [82], in 2019, Gonzalez-Villasana et al. [83] isolated these miRNAs in the exosomes from serum of both BC patients and healthy donors. However, this study confirmed significantly higher concentrations of exosomes in BC patients compared to healthy donors, supporting the hypothesis of an association between exosome concentration and the presence of BC.

In another study of 50 BC cases and 12 healthy controls, Eichelser and colleagues [84] reported that exosomal miR-101 and miR-372 were BC-specific, as confirmed by significantly higher serum levels in BC patients than in the control group. Moreover, Yoshikawa et al. [85] showed that plasma exosome-encapsulated miR-223-3p levels may be a useful preoperative biomarker to identify invasive lesions in patients diagnosed with DCIS by biopsy. In particular, exosomal miR-223-3p level was significantly increased in BC patients compared to healthy controls and showed a significant correlation with histological type, pT stage, pN stage, pathological stage, lymphatic invasion and nuclear grade.

In 2019, in order to investigate the enrichment of exosomal miRNAs in the pathogenesis of BC and DCIS, Ni et al. discovered an increase of exosomal miR-16 levels in plasma of BC and DCIS patients compared to healthy women, especially in cases of luminal tumors. Moreover, lower levels of exosomal miR-30b were associated with recurrence, and exosomal miR-93 was upregulated in DCIS patients [86].

In 2020, Zou et al. [87] focused on tumor regulation roles of members from the miR-532-502 cluster. They analyzed the expression patterns of miRNAs in the miR-532-502 cluster in about 800 plasma and serum samples from BC patients and healthy controls. They found that three miRNAs (miR-188-3p, miR-500a-5p, and miR-501-5p) in plasma and five miRNAs (miR-188-3p, miR-501-3p, miR-502-3p, miR-532-3p, and miR-532-5p) in serum were significantly up-regulated in BC patients. Similarly, Li et al. [88] identified five plasma miRNAs (let-7b-5p, miR-122-5p, miR-146b-5p, miR-210-3p and miR-215-5p) whose expression levels were significantly different in BC patients and controls. However, in plasma-derived exosomes, only miR-122-5p was consistently increased in BC patients. Moreover, evaluating the expression of 12 miRNAs from the miR-106a-363 cluster, the same authors [89] also identified three plasma-derived exosomal miRNAs (miR-106a-3p, miR-106a-5p, and miR-92a-2-5p) and three serum-derived exosomal miRNAs (miR-106a-5p, miR-19b-3p, and miR-92a-3p), whose levels were upregulated in BC patients.

Recently, Zou et al. [90] explored the expression of 12 miRNAs in 32 pairs of serum-derived exosome samples from BC patients and healthy controls, discovering 10 miRNAs (let-7b-5p, miR-106a-5p, miR-19a-3p, miR-19b-3p, miR-25-3p, miR-425-5p, miR-451a, miR-92a-3p, miR-93-5p, and miR-16-5p) to be consistently upregulated in serum-derived exosomes in BC patients compared to controls. Furthermore, Wang et al. [91] found that total circulating miR-188-5p was abnormally elevated in BC patients, compared to both women with breast fibroadenoma and healthy subjects, and its level correlated with tumor stage. On the other hand, by analyzing exosomal miRNA only, miR-188-5p levels in the serum of BC patients were reduced compared to healthy controls. Moreover, they did not differ from patients with breast fibroadenoma.

Interestingly, Li and colleagues [92] demonstrated that serum exosomal miR-148a levels were significantly downregulated in patients with BC as compared to healthy patients with benign breast tumors. Besides, the downregulation of serum exosomal miR-148a is closely associated with staging at diagnosis and disease relapse, indicating that it might be a promising non-invasive diagnostic and prognostic biomarker for BC. On the other hand, Rodriguez-Martinez and colleagues [93] investigated the use of serum exosomal miRNAs as diagnostic biomarkers in 53 patients initially diagnosed with locally advanced BC. They discovered that before neoadjuvant therapy, exosomal miR-21 and miR-105 expression levels were higher in metastatic versus non-metastatic patients and healthy controls. Based on these results, the authors suggested adding miR-21 and miR-105 analysis to mammogram tests, in order to identify those patients with metastatic disease who are misdiagnosed as non-metastatic by current clinical methods.

Some of the above studies demonstrated a different miRNAs’ expression profile according to tumor subtype. More specifically, exosomal miR-373 serum level was more elevated in case of TNBCs than in luminal cancers or unaffected patients [84]. Likewise, higher levels of serum exosomal miR-222 were observed in basal-like and in luminal B versus luminal A tumor subtypes [93]. In addition, the analysis of exosomal miRNA from BC cell lines using a Surface-Enhanced Raman Scattering (SERS) sensor confirmed a significantly higher expression of miR-21 in luminal and TNBCs compared to HER2-positive BCs, as previously reported [94]. On the other hand, miR-222 was detected in TNBCs and high level of miR-200c was observed in HER2-positive BCs [95]. Moreover, for the purpose of identifying particular miRNA signatures in exosomes derived from plasma of HER2-positive BC and TNBC patients, Stevic and colleagues [96] discovered 10 miRNAs (miR-27a/b, miR-30c, miR-150, miR-152, miR-199a-3p, miR-340, miR-376a, miR-410, and miR-598) in the entire cohort of BC patients, 13 miRNAs (miR-27a/b, miR-30c, miR-150, miR-152, miR-199a-3p, miR-328, miR-340, miR-365, miR-410, miR-422a, miR-598, and miR-628) in a subgroup of 211 HER2-positive BCs, and 17 miRNAs (miR-27b, miR-30c, miR-128a, miR-145, miR-150, miR-152, miR-199a-3p, miR-324-3p, miR-335, miR-340, miR-376a/c, miR-382, miR-410, miR-423-5p, miR-433, and miR-598) in the subgroup of 224 TNBC, significantly deregulated.

Based on a case-control study of 69 BC patients vs. 40 healthy controls, interestingly, Hirschfeld and colleagues [97] have recently identified a specific panel of four urinary microRNA (miR-424, miR-423, miR-660, and let7-i) as a highly specific combinatory biomarker tool discriminating BC patients from healthy controls, with 98.6% sensitivity and 100% specificity.

Studies of exosomal miRNA detected in serum and plasma of BC patients and potentially useful for early diagnosis are summarized in Table 1. To date, numerous studies on exosomal miRNAs linked to tumors, and BC in particular, have been published. This number is destined to increase, given the growing curiosity and attention toward this new potential diagnostic and prognostic tool. However, further research is needed in order to identify the most focused and promising set of miRNAs. 

## 6. Exosomal MicroRNAs for Early Diagnosis in High-Risk Patients

For women in Western countries, the average lifetime risk to develop BC is approximately 13% (i.e., 1 in 8 to 1 in 7 women) [98,99]. For the purpose of screening recommendations, women are stratified into two categories: average risk and increased risk. According to National Comprehensive Cancer Network (NCCN) guidelines [100], increased risk is determined by one of the following factors: (1) prior history of BC; (2) age ≥ 35 years with a 5-year risk of invasive BC ≥ 1.7% (Gail model [101]); (3) lifetime risk of BC >20% based on a history of mammary lesions of uncertain malignant potential (lobular carcinoma in situ, atypical ductal or lobular hyperplasia); (4) lifetime risk of BC > 20% based on models dependent on family history; (5) thoracic irradiation before 30 years of age (e.g., mantle irradiation) and (6) known genetic predisposition to BC.

Women with an increased risk to develop BC undergo more intensive screening that includes semestral to annual clinical examination and periodic breast imaging, often starting at an earlier age than the rest of the population [99,102]. Screening programs are associated with the risk of false-positive results and consequent over-diagnosis and overtreatments. In addition, they are associated with the possibility of false-negative results [103], which must be taken into account in case of elevated BC risk, especially in the presence of familial or known genetic predisposition to BC. These aspects contribute to increasing anxiety and distress in high-risk women, who would benefit from less invasive and more accurate diagnostic strategies.

Considering hereditary BC tumors, most cases are due to germline mutations in the breast cancer genes *BRCA1/2*. In this regard, the frequency of *BRCA1/2* pathogenic variants in the population has been estimated to be one in 400–500 [104,105]. Women’s risk of developing BC is 46–87% in *BRCA1* mutational carriers and 38–84% in *BRCA2* mutational carriers before 70 years of age [106]. While *BRCA2*-associated BCs are quite similar to sporadic tumors, *BRCA1*-related BCs have a triple-negative phenotype in 68% of cases [107]. In women undergoing regular BC screening, TNBC usually presents as an interval cancer (between two mammograms). Even in cases of more intensive screening programs, patients often present with palpable tumor because of the high proliferation index of TNBC [108]. Moreover, despite their large size at diagnosis, TNBCs can be occult on initial mammography [109]. Therefore, a more sensitive and less invasive screening method that would allow for early diagnosis is sorely needed in this setting. The analysis of exosomal miRNAs gathered through liquid biopsy could be a promising tool. Given the high percentage of TNBC in *BRCA1* mutational carriers, the isolation of TNBC-specific exosomal miRNAs might be a valid approach.

As mentioned above, serum and/or plasmatic levels of exosomal-annexin A2 [64] and of several exosomal miRNAs have been shown to be altered in TNBC [84,92,95]. Some studies on tumor specimens and cell lines discovered different miRNA selectively expressed in TNBC and possibly detectable through liquid biopsy. In detail, miR-155 is upregulated in tumors with *BRCA1* loss of function [110]. Moreover, miR-210 is upregulated in TNBC compared to luminal BC [111,112], and it is highly expressed in familial BCs compared to non-familial ones [113]. Besides, miR-221/222 are basal-like specific miRNAs that promote cell migration and invasion [114], while miR-34a levels are more than three-fold lower in TNBC cell lines than in normal and HER2-positive cell lines [115].

Until now, the potential role of exosomal miRNAs in early BC diagnosis in high-risk patients has not yet been investigated. Nevertheless, in *BRCA1* mutation carriers, a panel of multiple TNBC-specific exosomal miRNAs on serum or plasma performed more frequently than breast imaging might help to anticipate the TNBC diagnosis. This would be instrumental in reducing related morbidity and mortality and should therefore be further studied in this setting.

## 7. Conclusions

The clinical benefit of exosomes as a diagnostic biomarker in BC requires further data from large clinical trials, as most of the existing evidence is based on small cohort studies. Periodic assessment of several BC-specific exosomal miRNAs detected in the studies mentioned above may be useful to anticipate radiological diagnosis, particularly in women at increased risk of developing BC, thus overcoming the limits of present screening programs.

## Figures and Tables

**Figure 1 biomedicines-09-00312-f001:**
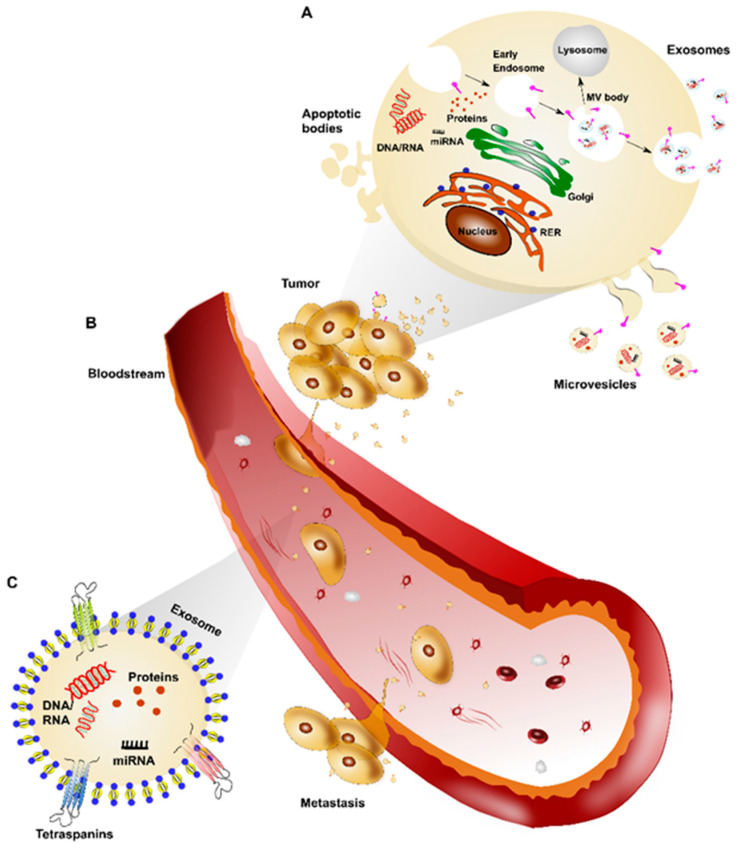
Extracellular vesicle (EV) biogenesis and role in tumor. (**A**). Tumor cells can release EVs involved in cell-to-cell communication; these are classified as exosomes, microvesicles and apoptotic bodies based on their origin and size. (**B**) Exosomes released by tumor cells reach the bloodstream and play an important role in metastasis development. (**C**) Exosomes are small lipid bilayer-enclosed vesicles, characterized by the presence of transmembrane tetraspanin proteins and a content of RNA, DNA, miRNA as well as proteins.

**Table 1 biomedicines-09-00312-t001:** Exosomal miRNA detected in serum or plasma of BC patients that could be useful for early diagnosis.

miRNA	Special Characteristics	References
miR-21 and mi-R1246	Significantly high in BC although ubiquitous in human exosomes	Hannafon et al. [78]
miR-145, miR-155 and miR-382	Significantly high in BC although ubiquitous in human exosomes	Gonzalez-Villasana et al. [83]
miR-101 and miR-372	Significantly high in BC	Eichelser et al. [84]
miR-223-3p	Significantly high in BC	Yoshikawa et al. [85]
miR-16	Significantly high in BC, especially if estrogen-positive	Ni et al. [86]
miR-93	Significantly high in DCIS	Ni et al. [86]
miR-188-3p, miR-500a-5p and miR-502-3p (miR-532-502 cluster)	Significantly high in BC	Zou et al. [87]
miR-122-5p	Significantly high in BC	Li et al. [88]
miR-106a-3p, miR-106a-5p, miR-92a-2-5p, miR-19b-3p and miR-92a-3p (miR-106a-363 cluster)	Significantly high in BC	Li et al. [89]
let-7b-5p, miR-106a-5p, miR-19a-3p, miR-19b-3p, miR-25-3p, miR-425-5p, miR-451a, miR-92a-3p, miR-93-5p and miR-16-5p	Significantly high in BC	Zou et al. [90]
miR-148a	Significantly downregulated in BC	Li et al. [92]
miR-27a/b, miR-30c, miR-150, miR-152, miR-199a-3p, miR-340, miR-376a, miR-410 and miR-598	Significantly deregulated in BC	Stevic et al. [96]
miR-21 and miR-105	Significantly high in metastatic vs. localized BC	Rodriguez-Martinez et al. [93]
miR-373	Significantly high in TNBC	Eichelser et al. [84]
miR-222	Significantly high in TNBC and luminal B vs. luminal A BC	Rodriguez-Martinez et al. [93]
miR-27b, miR-30c, miR-128a, miR-145, miR-150, miR-152, miR-199a-3p, miR-324-3p, miR-335, miR-340, miR-376a/c, miR-382, miR-410, miR-423-5p, miR-433 and miR-598	Significantly deregulated in TNBC	Stevic et al. [96]
miR-27a/b, miR-30c, miR-150, miR-152, miR-199a-3p, miR-328, miR-340, miR-365, miR-410, miR-422a, miR-598 and miR-628	Significantly deregulated in HER2-positive BC	Stevic et al. [96]

## Data Availability

Not applicable.

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
