# Peer review of "The Role of Exosomes in Breast Cancer Diagnosis"

_biomedicines, 2021, doi:10.3390/biomedicines9030312_

Round 1

Reviewer 1 Report

The authors have submitted a very nice review covering the important topic of exosomes in breast cancer diagnosis. They have summarized critical recent studies and highlighted the significant potential for the use of extracellular vesicle analysis in this regard. Consideration of the comments below will help to improve the final version for readers of the manuscript.

The authors define extracellular vesicles as exosomes, microvesicles and apoptotic bodies, however the review focuses almost exclusively on exosomes (titles for sections 2-6 all contain "exosome") and this is stated by the authors (lines 30, 155). The current title does not reflect this and indicates a broader discussion of extracellular vesicles in diagnosis. An adjustment to the title to more accurately reflect the content for the reader should be considered.

The abstract is composed of a combination of lines directly copied or very minimally altered from the body of the manuscript (e.g. lines 53-55, 57-66, 89-90, 132-133, 155-157). The repeated statements were noticeable when reading. While the abstract will closely reflect statements and other discussion found in the manuscript, it should be unique.

Section 1: expanding upon what is mentioned for exosomes, microvesicles and apoptotic bodies regarding size range, cellular origin (endosomal/plasma membrane) and context regarding their biogenesis would be helpful.

Section 2: a number of significant statements are made that would benefit from additional detail. E.g. line 120: "...important in initiating or suppressing various signalling pathways in recipient cells, also responsible for metastasis and drug-resistance". Line 125: "...reach the bloodstream and play an important role in metastasis development". Although not the focus of the review, additional information regarding how exosomes affect "recipient cells", what these cells are, how EVs in the bloodstream affect metastasis, etc, would be helpful to the reader.

Additionally, it is stated that EVs are released from viable cells and can contain DNA. In what context do viable cells package DNA for export from the cell? How does this relate to the formation of apoptotic bodies? Etc.

Minor edit suggestions:

Line 46: Sentence may read better if "emerging" is changed to "emergence".

Line 60: "Liquid" should not be capitalized.

Line 162: ..tetraspanins "are" abundantly...

Line 203: "Luminal" should not be capitalized.

Line 249: “cases".

Line 338: “genes".

Line 344: “cases".

Line 354: "cellular lines" should be "cell lines".

Line 360: "HER2+" should be written as "HER2-positive" to be consistent with remainder of manuscript.

Reviewer 2 Report

The authors have submitted a very nice review covering the important topic of extracellular vesicles in breast cancer diagnosis. They have summarized critical recent studies for the use of exosomes analysis as a predictors of early cancer diagnosis.

This is a well-written and interesting review  thus I recommend this manuscript for Biomedicines publication.

Minor edit suggestions:

Line 60: "Liquid" should not be capitalized.

Line 162: tetraspanins "are"

Line 203: "Luminal" should not be capitalized.

Line 338: “genes".

Line 344: “cases".

Line 354: "cellular lines" may be "cell lines".   
